# Clamping Fatigue Properties of Shrink-Fit Holder

**DOI:** 10.3390/mi13081263

**Published:** 2022-08-05

**Authors:** Zhouyi Lai, Zhenyu Zhao, Ting Guo, Yuyin Luo, Houming Zhou, Changan Li

**Affiliations:** 1School of Sino-German Robotics, Shenzhen Institute of Information Technology, Shenzhen 518172, China; 2Shenzhen Chuangshi Machinery Co., Ltd., Shenzhen 518029, China; 3School of Mechanical Engineering, Xiangtan University, Xiangtan 411105, China

**Keywords:** shrink-fit holder, clamping fatigue, thermal stress, contact stress, time domain waveform

## Abstract

In order to explore the clamping fatigue properties of shrink-fit holders, ANSYS software was used in this study to analyze the thermal and contact stresses during the clamping process of the shrink-fit holder, and the fatigue analysis was performed by selecting the dangerous areas based on the two stresses. A numerical control shrink-fit holder clamping fatigue test device was manufactured, and the automatic clamping of the shrink-fit holder was executed in this study. After 500 clamping repetitions, a milling test was carried out on the shrink-fit bracket. By collecting the vibration signal of the workpiece during processing and measuring the change in the surface roughness of the workpiece, and then analyzing the change in the machining performance of the shrink-fit holder under different clamping times, we were able to compare and verify the accuracy of the finite element fatigue analysis.

## 1. Introduction

A shrink-fit holder is an important clamping element in high-speed machining systems and is an important component connecting the machine tool and cutter [1,2]. It is connected to the cutter only through interference fit, without any other parts; thus, the structure is simple and reliable. Moreover, the unique clamping form of a shrink-fit holder makes it simpler in structure, and it has higher clamping stability and rotation balance than a traditional tool system. At present, the research of the design of shrink-fit holders has mainly focused on their overall performance (stiffness, clamping force, surface quality, etc.) [3], efficient heating methods, mechanical properties, tool after assembly [4], etc. Vahid Ostad Ali Akbari [5] used finite element modeling to study various stiffness and damping behaviors in thermal shrinkage fit. However, there is little research on the clamping fatigue performance of shrink-fit holders in actual production and machining, due to changes in machining requirements or tool wear, because the holder needs to be frequently heated and cooled to change tools. When it reaches a certain number of clamping repetitions, the holder cracks due to fatigue. Under the action of high-speed centrifugal force, a tiny crack expands, weakening the clamping force of the holder to the tool, intensifying the vibration during milling, affecting the machining quality of the workpiece, even leading to the damage of the holder, and causing harm to the machine tool and the operators. The authors of [6,7] analyzed the dynamic characteristics and radial clamping rigidity of a shrink-fit holder when it was matched with the tool, while the authors of [8,9] performed an in-depth study on the mechanical characteristics and clamping temperature of shrink-fit holders. However, there is little research on its clamping fatigue. Therefore, it is necessary to study the clamping fatigue life of shrink-fit holders and replace the holders that cannot meet the machining requirements in advance.

In this study, the thermal and contact stresses in the clamping process were simulated by the finite element method with ANSYS software, and then the fatigue analysis was carried out in the dangerous areas selected by the two stresses. Based on the hot-air, low-temperature heating device, a numerical control clamping fatigue test device was trial-manufactured to execute automatic clamping of the shrink-fit holder. A milling test was carried out once every 500 clamping repetitions. Through analyzing the vibration signals of the workpieces and the changes in the surface roughness of the machined workpieces, the change in the shrink-fit holders’ milling performance under different clamping times was found, and the accuracy of the finite element simulation was verified.

## 2. Working Principle of Shrink-Fit Holder

Different shrink-fit holders have different materials, sizes, and interference magnitudes; thus, the analysis process for each is different. In this study, the model CS-12-110 shrink-fit holder produced by MST Corporation, Nara, Japan was selected as the experimental object. The material object and its structural diagram are shown in Figure 1.

At normal temperature, the tool cannot be directly set into the clamping opening. Therefore, the clamping opening needs to be heated by a heating device, and the diameter of the clamping opening should be larger than that of the tool to make the tool set [10]. Finally, after cooling, the high excess force formed between the clamping opening and the tool achieves stable clamping.

## 3. Finite Element Analysis of Clamping Fatigue of Shrink-Fit Holder

Fatigue refers to the phenomena of the fracture and failure of a structure under repeated alternating stress, even if the stress is less than its allowable stress. Fatigue is generally divided into high and low cycle fatigue by 10^4^ to 10^5^ times, respectively. Because the stress of a shrink-fit holder is high in the clamping process, which exceeds the elastic limit and causes plastic deformation, it belongs to low cycle fatigue.

The fatigue failure of a structure starts with cracks in some areas, usually located in the part with the highest concentration of stress. The fatigue strength of this part represents that of the whole component, and the part is called the danger zone. In modern engineering, ANSYS software by ANSYS, Inc., Pittsburgh, PA, USA is often used to analyze the fatigue life of workpieces. When using ANSYS to carry out fatigue analysis, the main data include: location, i.e., selection of danger zones; events, that is, application of load steps; and material parameters, mainly including S-N curve and other material performance parameters.

The stress of shrink-fit holders in the clamping process is mainly divided into two parts: one is the thermal stress generated by thermal expansion after heating, and the other is the contact stress generated by interference fit with the tool after cooling.

Because the overall structure and load distribution of the shrink-fit holder and tool meet the axisymmetric condition, in order to reduce the calculation amount, a simplified 2D finite element model was established, as shown in Figure 2. The grid was divided by a structural element PLANE182, and the model had 1774 elements and 2010 nodes [11]. The parameters required for analysis are shown in Table 1.

### 3.1. Thermal Stress Analysis of Shrink-Fit Holder

In this study, a hot-air low-temperature heating device was selected to heat the shrink-fit holder. It mainly conducted heat through heat conduction and heat convection.

Heat conduction follows Fourier’s law:(1)$$q^{1}=-k\frac{dT}{dx}$$

Here, $q^{1}$ is the heat flux ($W/m^{2}$); k is the thermal conductivity (W/(m·k)
$\frac{dT}{dx}\;$is the temperature gradient; the minus sign indicates that heat flows from high temperature to low temperature.

Heat convection can be described by Newton’s cooling equation:(2)$$q^{2}=h_{f}(T_{s}-T_{B})$$

Here,$\;h_{f}$ is the film coefficient; $T_{s\;}$is the solid surface temperature; $T_{B\;}$is the temperature of the surrounding fluid.

The thermal stress analysis in ANSYS is a coupling analysis of the thermodynamic module and structural mechanics module. First, calculate the temperature of the model in the thermodynamic module. Then, in the structural mechanics unit, apply the thermodynamic analysis result as a load to the model, and perform the static analysis [12,13].

According to [14], the film coefficient TCC is 40,000, and the convective heat transfer condition is forced convection of gas at 20 °C. The thermal unit PLANE55 was used to apply a temperature load to the clamping opening of the holder. After thermodynamic analysis, the steady-state temperature distribution diagram, as shown in Figure 3, was obtained.

It can be seen from the figure that the inner wall temperature of the shrink-fit holder after heating was about 290 °C, which was consistent with the actual measured clamping temperature of 268.5 °C [10]. After the steady-state temperature distribution result was obtained, the unit was converted into a structural unit PLANE182, and the thermodynamic analysis result was loaded into the model as a load to carry out static analysis. After solving, the thermal stress distribution diagram shown in Figure 4 was obtained.

### 3.2. Contact Stress Analysis of Shrink-Fit Holder

Because interference fit analysis is a highly nonlinear problem, if pure theory is used for analysis, it will be more complicated, so ANSYS can be used as an analysis tool to solve such problems.

According to the paper [15], the friction coefficient μ between the shrink-fit holder and tool is 0.2, and the magnitude of radial interference is 5 μm. Contact pairs are used to simulate interference fit between the shrink-fit holder and the tool [16]. Contact units TARGE170 and CONTACT 174 were selected to, respectively, simulate the tool contact surface with greater relative rigidity and the holder fitting surface with greater relative flexibility. Because the bottom of the shrink-fit holder should be fixed on the clamping seat during clamping, a full constraint setting was applied to the bottom surface. After solving, the contact stress distribution diagram between the shrink-fit holder and the tool was obtained, as shown in Figure 5.

### 3.3. Clamping Fatigue Analysis of Shrink-Fit Holder

The S-N curve is a material fatigue property curve obtained by repeatedly testing raw materials with a specific shape after being treated by a specified process, which shows the relationship between stress range S and material life N. Some of the values of the S-N curve of the shrink-fit holder material are shown in Table 2.

The points of maximum stress in thermal stress analysis and contact stress analysis were selected as the analysis positions. In addition, because the inner wall is the weak position of the tool rod, several points on the inner wall were sequentially selected as the analysis object. After the selection was completed, the contact and thermal stresses obtained from the above analysis are stored in the corresponding position and solved after input of material parameters.

Through analyzing the calculation results, we found that the fatigue life of the points on the inner wall of the shrink-fit holder was generally shorter, and the point with the largest contact stress had the shortest fatigue life. The calculation showed that the clamping life of this point was 3175 times, and the fatigue cumulative damage coefficients for 2000, 2500, and 3000 clamping repetitions were 0.62995, 0.78743, and 0.94492, respectively.

## 4. Test and Verification of Clamping Fatigue of the Shrink-Fit Holders

The fatigue analysis test method involves applying specific cyclic load to the research object to obtain fatigue life. The results obtained by the test method are accurate and reliable, but the labor and time costs are relatively high, so can generally be used as verification means.

We selected three new shrink-fit holders as test objects. We clamped the tools for many cycles, recorded the clamping times, and calculated the average value. The selected heating and cooling times of the clamping cycles were 150 and 450 s, respectively, and the cycle was completed every 10 min.

### 4.1. Numerical Control Shrink-Fit Holder Clamping Fatigue Test Device

A numerical control shrink-fit holder clamping fatigue test device, as shown in Figure 6, was trial-manufactured to perform automatic clamping of the shrink-fit holder and record the clamping times. The device mainly carries out automatic control on the whole device through a programmable controller, and the whole clamping process is as follows:

Firstly, control stepping motor 1 to drive the ball screw so that the three-jaw chuck holds the shrink-fit holder equipped with the cutter to reach the heating port of the hot-air heating device. After the holder reaches the designated position, start the hot-air, low-temperature heating device to heat the holder. After the heating time is reached, stepping motor 2 drives the ball screw to drive the pneumatic three-jaw chuck to clamp the tool and finish unloading and loading the tools. After the clamping is completed, the shrink-fit holder returns to the original position for air cooling. Once cooled to room temperature, the holder completes one cycle.

### 4.2. Milling Experiment Design

In the process of automatic clamping the shrink-fit holders, we carried out milling tests every 500 repetitions on the XD-40A numerical control milling machine, produced by Dalian Machine Tool Group, Dalian, China, with the highest rotating speed of the milling machine being 8000 r/min. We selected the hilt BT40-SLK12-75, produced by Japan MST Company, and aluminum alloy as the workpiece, and used the NI PXLE-1082 acquisition card, produced by National Instruments. The acquisition card tape automatically stored the hard disk, and through the computer display, the signal fluctuation collected in the milling process could be simply and clearly observed. A SAE30010 three-way piezoelectric acceleration sensor was selected. The measurement range (G) of the sensor was −50 to 50 the sensitivity (MV/g) was 50, and the frequency range (HZ) was 0.5~5000. During the test, we adhered the three-way piezoelectric acceleration sensor to the side of the workpieces, and collected the vibration signals during milling using the NI data acquisition card. The device diagram of the test is shown in Figure 7.

During cutting, the vibration amplitude is more stable when the parameters selected are in the cutting stability domain. According to [17,18], we selected the machining parameters, as shown in Table 3, in the stability domain.

### 4.3. Vibration Signal Analysis

MATLAB software by MathWorks, Inc, Neddick, ME, USA was selected to process the collected signals. After Fourier transform, the vibration signals of shrink-fit holders during machining were converted into vibration time domain signal diagrams in X, Y, and Z directions.

The analysis showed that the variation rules of vibration waveforms in the X and Y directions were basically the same; therefore, only the X direction was selected for analysis instead of repeating the analysis. The Z direction is the spindle direction. Because the rigidity of the spindle is large, the vibration amplitude in the Z direction is much smaller than that in the X and Y directions, so can be ignored [19]. Therefore, the X direction vibration time domain signal of the workpiece at 8000 rpm was finally selected as the analysis object. Because the cut-in part of the milling cutter has a large vibration, which produces large errors, and thus is not conducive to analysis, only the stable cutting part was intercepted. Figure 8 includes vibration waveform diagrams in the X direction of the workpieces, and Figure 9 includes vibration spectrum diagrams in the X direction of the workpieces.

As can be seen from Figure 8, the milling fluctuation of the shrink-fit holder was the smallest after the initial clamping. With the increase in clamping repetitions, the vibration of the shrink-fit holder increased with different intensities. The increase in amplitude from 1000 to 2500 clampings was relatively small. After 3000 clampings, the vibration of the shrink-fit holder obviously strengthened, and many vibration signals with amplitudes much larger than the average level appeared. When the number of clamping repetitions reached 4000, the vibration fluctuation of the shrink-fit holder reaches the maximum value. As can be seen from Figure 9, the frequencies of vibration mainly concentrated at 100, 300, and 600 HZ, of which 300 HZ had the largest amplitude. With the increase in clamping repetitions, the amplitude of each frequency increased. When the clamping times reached 3000, the amplitudes at 200, 500, and 800 HZ obviously enhanced.

The results of the analysis of the vibration waveform and spectrum showed that the amplitude of the shrink-fit holder slowly increased, and the overall increase was not large when the number of clampings was less than 2500. When it reached 3000 times, the amplitude s obviously enhanced, which indicated that the milling amplitude of the shrink-fit holder significantly increased and the milling performance rapidly decreased after 3000 clamping repetitions.

### 4.4. Analysis of Workpiece Surface Roughness

Roughness was measured by a TR200 hand-held roughness meter produced by Shandong Mountain Material Testing Instrument Company, Yantai, China. The measuring range of the meter was −20 to 20 μm, and the sensitivity was 0.01 μm. Five sets of data were measured under each set of parameters along the milling direction. After calculating the average values, the surface roughness of the workpieces under different clamping times of the shrink-fit holders was obtained, as shown in Figure 10.

As can be seen from the figure, with the increase in the clamping repetitions of the shrink-fit holders, the roughness of the machined surface of the workpieces showed an upward trend under different spindle rotation speeds, and the higher the rotation speed, the greater the increase. When the spindle speed was 8000 rpm, the surface roughness of the machined workpiece increased from Ra0.289 to Ra0.528 after 4000 clampings by lengthening the heated shrink-fit holder, which was 82.7% higher than that after the initial clamping. At 4000 and 6000 rpm, respectively, it increased from Ra0.631 and Ra0.408 to Ra0.752 and Ra0.578, an increase rate of 19.3% and 41.9%. Because he shrink-fit holders achieve high clamping accuracy by high interference force, repeated clamping reduces the clamping force, and the clamping force is further reduced under the centrifugal force of high rotating speed, the machining accuracy of the workpiece at 8000 rpm decreased even more seriously. At the same rotating speed, the workpiece surface roughness slightly changed from the first clamping to the clamping of 2000 times, and increased after clamping 2500 times. The workpiece surface roughness reached the maximum value at each rotating speed when the number of clamping times was 4000.

Combined with the above vibration analysis, we found that after clamping 2500 times, the machining performance of the shrink-fit holders significantly decreased. The fatigue cumulative damage coefficient of clamping 2500 times in the above finite element method reached 0.78743. In addition, the clamping force of the shrink-fit holder was greatly reduced under the action of centrifugal force at high rotating speed [20]. Therefore, it agrees with the finite element analysis result.

### 4.5. Clamping Fatigue Test of Shrink-Fit Holders

After 3700, 3900, and 4000 repetitions o clamping, cracks visible to the naked eye appeared on the three holders. The cracks on the surface of the third holder, after 4000 times of clamping, are shown in Figure 11.

Because it is difficult to monitor cracks in real time, and internal cracks cannot be detected at the time they are initiated, there is a lag between initiation and when cracks are visible to the naked eye. The number of clamping times when the actual observed cracks appear is larger than the number of times when cracks are initiated. Therefore, the test results are larger than the finite element simulation analysis results, which we expected.

Considering the above tests, we concluded that the shrink-fit holder is no longer suitable for high-speed milling after clamping 2500 times and needs to be replaced after clamping 3500 times.

## 5. Conclusions

The clamping fatigue of the shrink-fit holder was analyzed by using finite element software, ANSYS. A numerical control shrink-fit holder clamping fatigue test device was manufactured to automatically finish clamping, and milling tests were carried out on the shrink-fit holders after clamping for different times. The results showed that:(1)The thermal stress of the shrink-fit holder after heating and the contact stress with the tool after cooling were analyzed by ANSYS software, and the dangerous area was selected to carry out fatigue analysis, which showed that the clamping fatigue life of the shrink-fit holder is 3275 times.(2)Through analyzing the vibration signals in the milling process, we found that the amplitude slowly increases with the increase in clamping repetitions, so the vibration was relatively stable within 2500 clampings. After 2500 clampings, the amplitude obviously increased, and there were many messy signals, which indicated that the stability of the shrink-fit holders significantly decreases after 2500 clamping repetitions.(3)By analyzing the surface roughness of the machined workpieces, we found that the overall roughness increases with the increase in clamping times, especially at 8000 rpm. After 2500 clampings, the increase in workpiece surface roughness was obvious. When the number of clamping times reached 4000, the increasing workpiece surface roughness reached the maximum.(4)After 3700, 3900, and 4000 repetitions of clamping, cracks visible to the naked eye appeared on the surface of the three holders. The shrink-fit holder is no longer suitable for high-speed milling after 2500 times of clamping and needs to be replaced after 3500 times of clamping.

## Figures and Tables

**Figure 1 micromachines-13-01263-f001:**
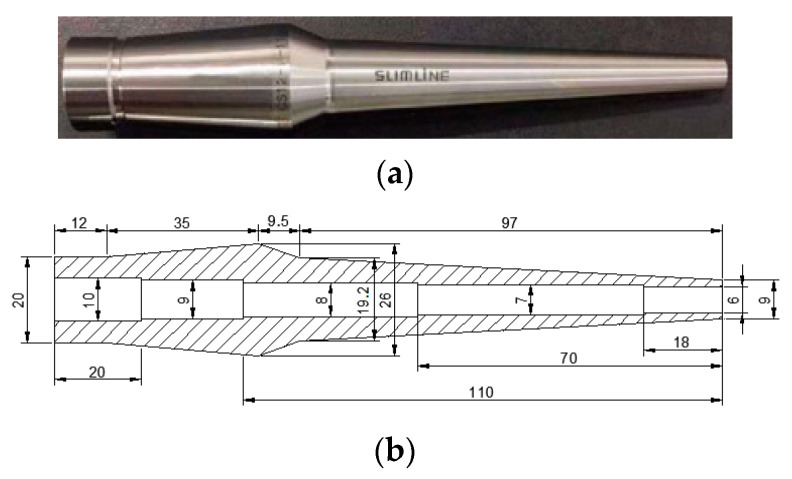
Shrink-fit holder. (**a**) Material object; (**b**) structural diagram.

**Figure 2 micromachines-13-01263-f002:**
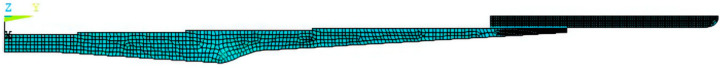
Finite element model of the shrink-fit holder and tool.

**Figure 3 micromachines-13-01263-f003:**
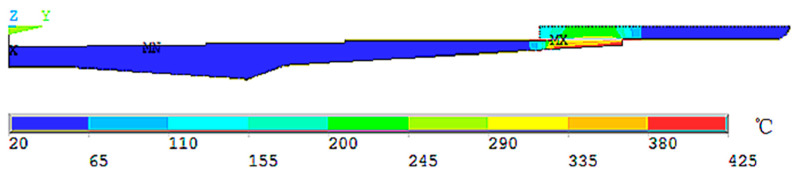
Steady-state temperature distribution diagram.

**Figure 4 micromachines-13-01263-f004:**
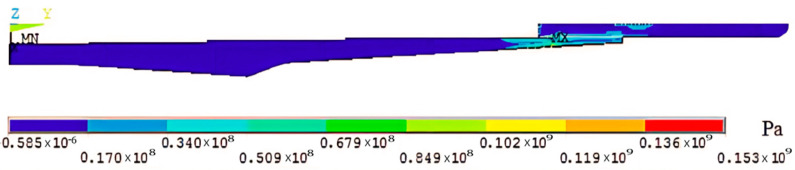
Thermal stress distribution diagram.

**Figure 5 micromachines-13-01263-f005:**
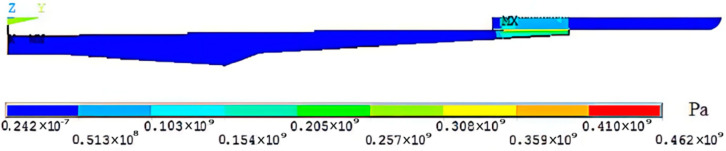
Contact stress distribution diagram.

**Figure 6 micromachines-13-01263-f006:**
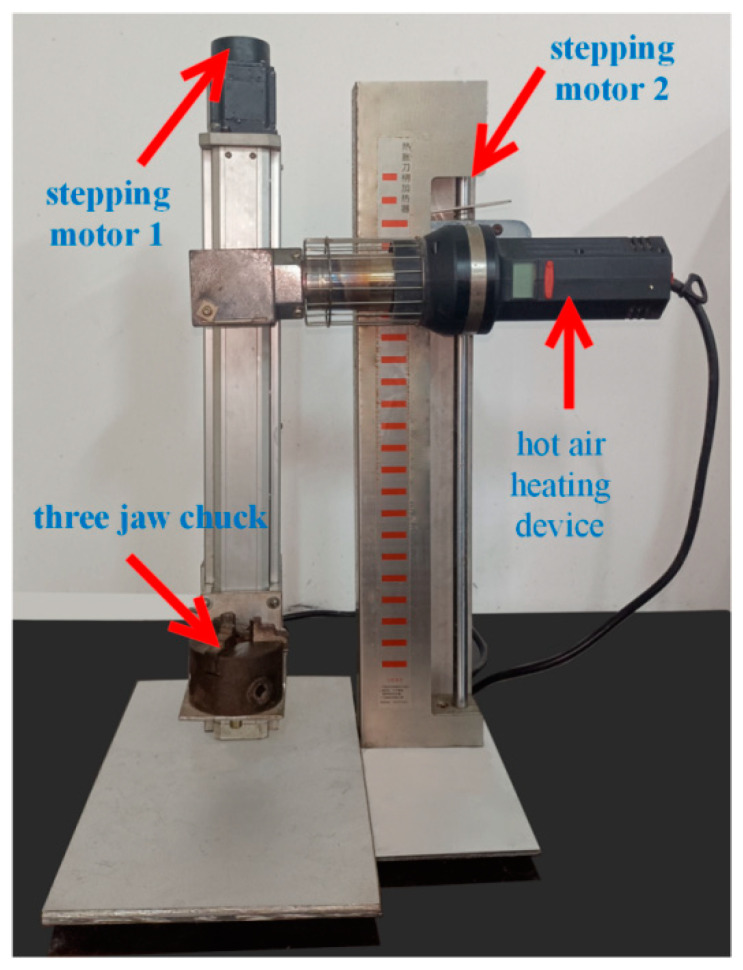
Numerical control shrink-fit holder clamping fatigue test device.

**Figure 7 micromachines-13-01263-f007:**
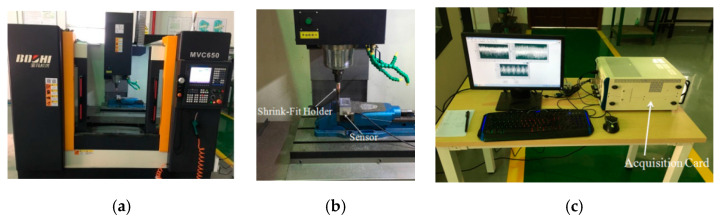
Milling test device diagram. (**a**) Numerical control machining center; (**b**) sensor; (**c**) acquisition card.

**Figure 8 micromachines-13-01263-f008:**
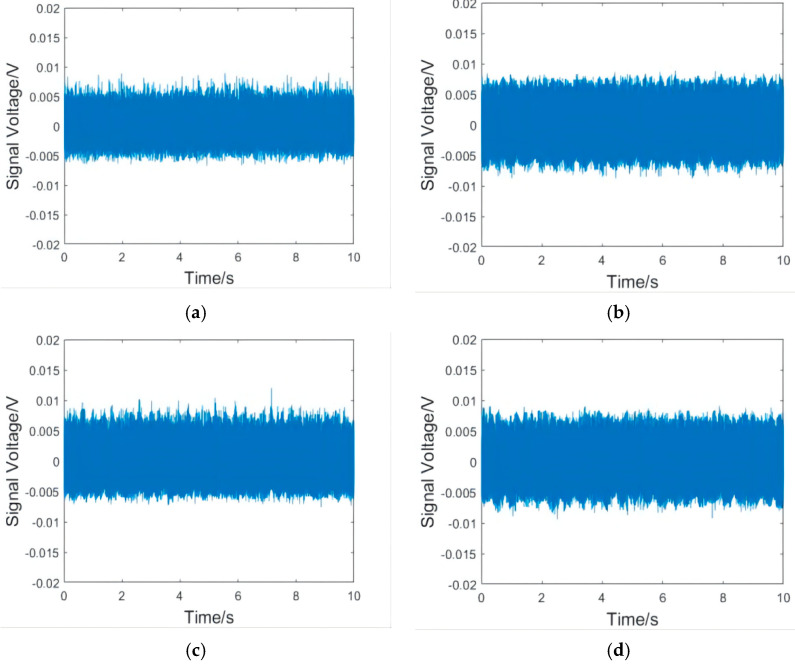
Vibration waveform in the X direction of the workpiece at 8000 rpm. (**a**) After the initial clamping; and after (**b**) 1000, (**c**) 1500, (**d**) 2000, (**e**) 2500, (**f**) 3000, (**g**) 3500, and (**h**) 4000 clamping repetitions.

**Figure 9 micromachines-13-01263-f009:**
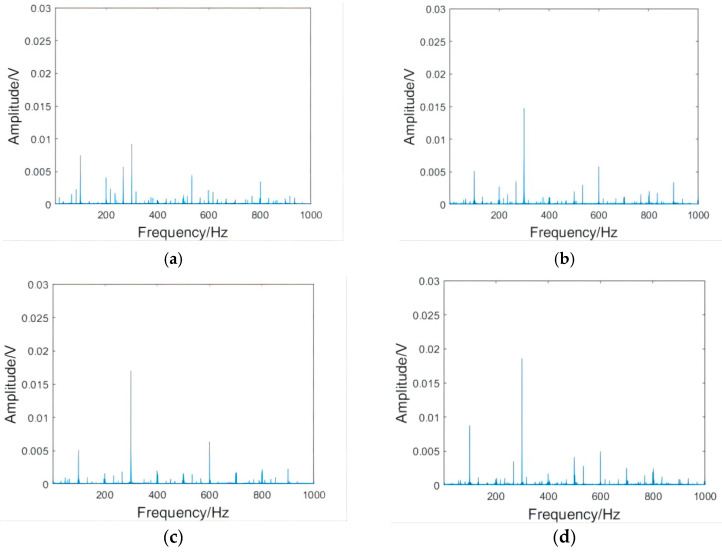
Vibration spectrum in the X direction of the workpiece at 8000 rpm. (**a**) After the initial clamping; and after (**b**) 1000, (**c**) 1500, (**d**) 2000, (**e**) 2500, (**f**) 3000, (**g**) 3500, and (**h**) 4000 clamping repetitions.

**Figure 10 micromachines-13-01263-f010:**
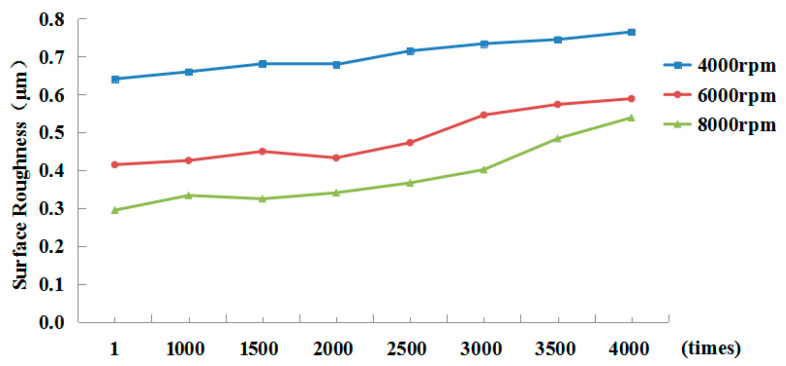
Workpiece surface roughness.

**Figure 11 micromachines-13-01263-f011:**
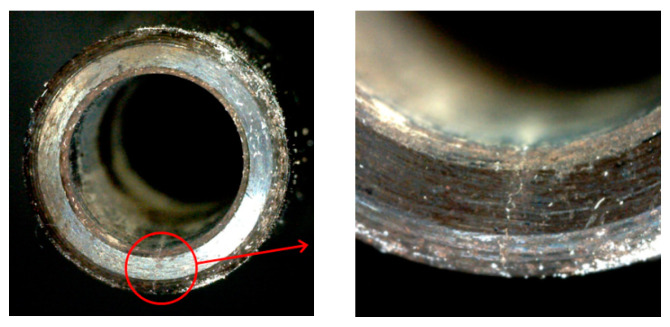
Surface crack of shrink-fit holder.

**Table 1 micromachines-13-01263-t001:** Parameters of the shrink-fit holder the tool.

	Shrink-Fit Holder	Tool
Elasticity modulus (GPa)	200	640
Poisson’s ratio	0.3	0.22
Density (kg/m^2^)	7800	15,000
Yield strength (MPa)	530	5460
Coefficient of thermal expansion (1/°C)	1.172 × 10^−5^	4.50 × 10^−6^
Specific heat capacity J/(kg·K)	504	200
Thermal conductivity w/(m·k)	67.2	80
Materials	Special stainless steel for hot loading	Superfine alloy matrix and TiAlN coatin
Model	CS-12-110	GM-4E-D6.0
Heat-resistance temperature	720 °C	-

**Table 2 micromachines-13-01263-t002:** S-N curve table.

N	10	20	50	100	200	1000
S	4 × 10^9^	2.827 × 10^9^	1.89 × 10^9^	1.413 × 10^9^	1.069 × 10^9^	4.41 × 10^8^
N	10,000	20,000	1 × 10^5^	2 × 10^5^	1 × 10^6^	
S	2.62 × 10^8^	2.14 × 10^8^	1.38 × 10^8^	1.14 × 10^8^	8.62 × 10^7^	

**Table 3 micromachines-13-01263-t003:** Milling test machining parameters.

Machining Method	Plane Machining
Speed n	4000–8000 (rpm)
Axial depth Ad	0.2 (mm)
Radial depth Rd	0.5 (mm)
Feed per tooth fz	0.3 (mm/z)

## Data Availability

Not applicable.

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
