# Peer review of "Clamping Fatigue Properties of Shrink-Fit Holder"

_micromachines, 2022, doi:10.3390/mi13081263_

Round 1

Reviewer 1 Report

1.     The methods employed in this paper were widely used and the object was a commercial product. It seems that the paper just uses conventional methods for analyzing the clamping fatigue properties of shrink-fit holder. There is also no optimization on the structures of the product for improving the based performance. What are the objectives and innovations of this paper?

2.     The current research status on the clamping fatigue properties of shrink-fit holder were not investigated in introduction.

3.     More references have to be added.

4.     How to mount the tool onto the holder in the experimental section.

5.     The figures 3, 4, 5, 8, 9 have to be clearer and larger for reading.

Author Response

Dear Editors and Reviewers:

Thank you for your letter and for the reviewers' comments concerning our manuscript entitled ”Research on Clamping Fatigue Properties of Shrink-Fit Holder". Those comments are all valuable and very heIpfuI for revising and improving our paper, as welI as the important guiding significance to our researches. We have studied cooments carefully and have made correction which we hope meet with approval. Revised portion are marked in red in the paper. The main corrections in the paper and the responds to the reviewers comments are as flowing:

Responds to the reviewers comments:

  1. Response to comment: What are the objectives and innovations of this paper?

   Response: In the actual production and processing process, due to the change of processing requirements or tool wear, the lengthened heat shrinkable tool rod needs to be heated and cooled frequently to change the tool. Repeated tool loading will cause fatigue damage of the tool rod, affect the quality of workpiece processing, and even lead to the destruction of the tool rod, causing harm to the machine tool and operators.  This topic for extended hot shrinkage clamping tool rod fatigue characteristics, the integrated use of the theory of fatigue damage, cutting theory and elastic-plastic mechanics, thermodynamics, the finite element analysis and experimental verification method, comprehensively analyzed the different number of times of the clamping extended the influence of thermal shrinkage tool rod milling performance, extended hot shrinkage is given reasonable service life of the,  It has a certain reference significance to the timely replacement of the cutter rod in the actual production and processing, and avoids unnecessary damage caused by the damage of the cutter rod. 

  1. Response to comment:The current research status on the clamping fatigue properties of shrink-fit holderwere not investigated in introduction.

   Response: The research status of shrinkage combined gripper clamping has been introduced in line 27-33 of the paper. 

  1. Response to comment:More references have to be added.

 Response: Seven references have been added as requested, most of which are from MDPI .

  1. Response to comment: How to mount the tool onto the holder in the experimental section.

Response: This issue has been explained and annotated in line 171-178 as requested .

  1. Response to comment: The figures 3, 4, 5, 8, 9 have to be clearer and larger for reading.

 Response: Figures 3, 4, 5, 8, 9 have been enlarged and clarified as required.

Reviewer 2 Report

The paper deals with the Research on Clamping Fatigue Properties of Shrink-Fit Holder.

The reviewer believes that the paper addresses a burning issue in manufacturing engineering, 

such as improving the quality of machining processing.

Because this approach is innovative,

the reviewer suggests the paper for publication.

But, there are needs major revision notes 

that are highlighted for the best presentation of the paper.

Comment 1

Extended text editing

Line 4

518029,China 

The authors should replace (insert a space)

518029, China 

Line 5

518029,China

The authors should replace (insert a space)

518029, China

Line 13

It is not so good to use the word "we". The authors must rephrase.

Line 18

shrink-fit holder ;

The authors should replace (delete a space)

shrink-fit holder;

Comment 2

Line 21

0. Introduction

The authors must format the paper according to the journal's instructions.

1. Introduction

The authors must renumber the othes sections.

Line 24

only   through

The authors should replace (delete a space)

only through

Comment 3

Line 23

tool and cutter [1~2]. 

The authors should replace

tool and cutter [1-2]. 

Line 30

of clamping times , the holder

The authors should replace

of clamping times, the holder

Line 32

force of   the holder

The authors should replace

force of the holder 

Line 34

The papers[3~4] analyze 

The authors should replace

The papers [3-4] analyze 

Line 36

the papers[5~6] have

The authors should replace

the papers [5-6] have

Line 44

a numerical control   clamping fatigue

The authors should replace

a numerical control clamping fatigue

Line 59

the stable clamping . 

The authors should replace (delete a space)

the stable clamping. 

Line 77

load steps;   material parame-

The authors should replace (delete a space)

load steps; material parame-

Comment 4

Figures 2, 8 and 9

The Figure must be accompanied on the same page as the Figure's title.

Comment 5

The authors must give more details for the FEM simulation (number of the elements, type of the elements, boundary conditions).

Comment 6

Table 1

Elasticity   modulus(GPa) 

The authors should replace

Elasticity modulus (GPa) 

The authors must add (Table 1) the materials of the shrink-fit holder and the tool. 

The authors must give more details for the mechanical properties (from supplier or authors experiments?).

Comment 7

Line 89

a hot-air low-temperature heating device is selected

The authors must give more details for the heating device (type, model) 

Line 97

Here,â„Ž? is film

The authors should replace (insert a space)

Here, â„Ž? is film

Line 102

model   and do the static analysis.

The authors should replace

model and do the static analysis.

Line 103

According to the paper[7], the film

The authors should replace (no upper case)

According to the paper [7], the film

Line 105

of the holder. after thermodynamic

The authors should replace

of the holder. After thermodynamic

Line 112

of 268.5℃[6]. After the

The authors should replace (no upper case)

of 268.5℃ [6]. After the

Line 118

Analysis of    the Shrink-Fit Holder

The authors should replace

Analysis of the Shrink-Fit Holder

Line 122

According to the paper [8], the friction

The authors should replace (no upper case)

According to the paper [8], the friction

Line 123

is 5 μ m. Contact

The authors should replace

is 5 μm. Contact

Line 142

selected   together as the

The authors should replace

selected together as the

Line 160

6 is   trial-manufactured

The authors should replace

6 is trial-manufactured

Comment 8

The authors must add a Figure (after the Figure 6) with the real Numerical Control Shrink-Fit Holder Clamping Fatigue Test Device.

Comment 9

Line 174

shrink-fit holders , carry

The authors should replace

shrink-fit holders, carry

Line 177

Select the hilt BT40-SLK12-75   produced by Japan MST Company, and   aluminum alloy

The authors should replace (delete the spaces)

Select the hilt BT40-SLK12-75 produced by Japan MST Company, and aluminum alloy

Line 178

the three-way piezoelectric acceleration sensor

The authors must give more details for the three-way piezoelectric acceleration sensor (type, model) and the acquisition card (type, model).

Line 183

According to the papers[9~10], select in the 

The authors should replace

According to the papers [9-10], select in the 

Line 184

shown in Table 3 .

The authors should replace

shown in Table 3.

Line 201

X Direction of   the Workpiece

The authors should replace

X Direction of the Workpiece

Line 215

shrink-fit holder   increases slowly

The authors should replace

shrink-fit holder increases slowly

Line 216

is less than 2,500 .

The authors should replace

is less than 2,500.

Line 220

by TR200 hand-held roughness meter, and

The authors must give more details for the TR200 hand-held roughness meter (type, model).

Line 227

fit holders , the roughnesses

The authors should replace

fit holders, the roughnesses 

The authors must give more details for the Surface Roughness (Ra or Rz or....).

Comment 10

Line 243

The authors must give more details how this value (0.78743) is calculated.

Comment 11

The authors did not comment inside the paper the ref. [11] - [14].

The authors must comment all the references in the paper.

Comment 12

Changes the References section format.

According to the journal's instructions:

1. Author 1, A.B.; Author 2, C.D. Title of the article. Abbreviated Journal Name Year, Volume, page range.

The authors must correct the References according to the journal's instructions.

The authors must remove the symbols [J] and [M].

Increase the number of the reference papers including (primarily) from Micromachines.

The authors use 0 paper from Micromachines journal / 0 papers from MDPI Journals / 14 papers from journals (References)

Τhe number for papers from MDPI journals

is considered insufficient (in reviewer's opinion).

Author Response

Dear Editors and Reviewers:

Thank you for your letter and for the reviewers' comments concerning our manuscript entitled ”Research on Clamping Fatigue Properties of Shrink-Fit Holder". Those comments are all valuable and very heIpfuI for revising and improving our paper, as welI as the important guiding significance to our researches. We have studied cooments carefully and have made correction which we hope meet with approval. Revised portion are marked in red in the paper. The main corrections in the paper and the responds to the reviewers comments are as flowing:

Responds to the reviewers comments:

  1. Response to comment: Extended text editing.

 Response: As required, the text editing questions raised by reviewers have been modified one by one, and the inappropriate use of WE in line 13 of the paper has been modified. Change to ”After 500 times of clamping, milling test was carried out on the shrink-fit bracket”.

  1. Response to comment: The authors must format the paper according to the journal's instructions.

   Response: The paper format has been arranged according to the journal's instructions, and the subsequent arrangement has been changed. 

  1. Response to comment: Tool and cutter [1-2],et al.

   Response: As required, the text editing questions raised by reviewers have been modified one by one.

  1. Response to comment: Figures 2, 8 and9. The Figure must be accompanied on the same page as the Figure's title.

   Response: The diagram and its title have been edited on the same page.

  1. Response to comment: The authors must give more details for the FEM simulation (number of the elements, type of the elements, boundary conditions).

 Response: For finite element simulations, the authors have added more details (including number of elements, element types, boundary conditions) in line 85 of the paper. The boundary conditions are mainly placed on lines 126-128.

  1. Response to comment:The authors must add (Table 1) the materials of the shrink-fit holder and the tool.

 Response: Materials and partial properties of shrink-fit brackets and tools have been added in Table 1.

  1. Responseto comment: A hot-air low-temperature heating device is selected, et al.

   Response: The model and other details of the heating device have been given at lines 255-257. As required, the text editing questions raised by reviewers have been modified one by one.

  1. Response to comment: The authors must add a Figure (after the Figure 6) with the real Numerical Control Shrink-Fit Holder Clamping Fatigue Test Device.

   Response: The actual numerical control shrinkage with tool holder clamping fatigue test device diagram has been added.

  1. Response to comment: The authors must give more details for the three-way piezoelectric acceleration sensor.

   Response: As required, the text editing questions raised by reviewers have been modified one by one. The three-way piezoelectric acceleration sensor has been introduced at lines 190-192, more details on surface roughness are given at LINES 266-270.

  1. Response to comment: The authors must give more details how this value (0.78743) is calculated.

   Response: More details on how 0.78743 is calculated are given in line 143-149 of the article.

  1. Response to comment: The authors did not comment inside the paper the ref. [11] - [14].

   Response: All references in the paper have been annotated.

  1. Response to comment: Changes the References section format.

 Response: References have been modified according to the journal description, symbols [J] and [M] have been removed, and the number of reference papers has been increased, mainly from MDPI .

Thank you again for your valuable modification suggestions!  At the same time, I hope you can inform me in time if you find any shortcomings again in the review process, and I hope the article can be published as soon as possible. 

Round 2

Reviewer 1 Report

I suggested to accept this manuscript in present form.

Reviewer 2 Report

Comment 1

Extended text editing

Lines 8 - 9

The authors should delete the gap between two lines.

Line 12

article.After

The authors should replace (insert a space)

article. After

Line 29

quality, etc.)[3], efficient 

The authors should replace (insert a space)

quality, etc.) [3], efficient 

Line 30

assembly[4], etc. Vahid Ostad Ali Akbari[5] used finite

The authors should replace (insert a space)

assembly [4], etc. Vahid Ostad Ali Akbari et al. [5] used finite

Line 39

The papers[6-7] analyze

The authors should replace (insert a space)

The papers [6-7] analyze

Line 41

while the papers[8-9] have made 

The authors should replace (insert a space)

while the papers [8-9] have made 

Line 62

the tool set[10]. 

The authors should replace (insert a space)

the tool set [10].

Line 90

and 2010 nodes[11]. Parame-

The authors should replace (insert a space)

and 2010 nodes [11]. Parame-

Line 109

the static analysis[12-13].

The authors should replace (insert a space)

the static analysis [12-13].

Line 110

According to the paper[14], the film

The authors should replace (insert a space)

According to the paper[14], the film

Line 119

268.5℃[15]. After 

The authors should replace (insert a space)

268.5℃ [15]. After 

Line 129

According to the paper[16], the

The authors should replace (insert a space)

According to the paper [16], the

Line 131

and the tool[17]. Contact units

The authors should replace (insert a space)

and the tool [17]. Contact units

Line 198

According to the papers[18-19], select

The authors should replace (insert a space)

According to the papers [18-19], select

Line 209

ignored[20]. Therefore,

The authors should replace (insert a space)

ignored [20]. Therefore,

Line 262

rotating speed[21]. Therefore,

The authors should replace (insert a space)

rotating speed [21]. Therefore,

Comment 2

The authors must format the paper according to the journal's instructions.

The Lines 41 - 45 and the Lines 46 - 54: have not the same text format.

Comment 3

Changes the References section format.

According to the journal's instructions:

1. Author 1, A.B.; Author 2, C.D. Title of the article. Abbreviated Journal Name Year, Volume, page range.

The authors must correct the References according to the journal's instructions.

Ref 17: Delete the [J]